# The Urinary Microbiome and Bladder Cancer

**DOI:** 10.3390/life13030812

**Published:** 2023-03-17

**Authors:** Nassib Abou Heidar, Tariq A. Bhat, Usma Shabir, Ahmed A. Hussein

**Affiliations:** 1Department of Urology, Roswell Park Comprehensive Cancer Center, Buffalo, NY 14203, USA; 2Department of Urology, Cairo University, Giza 12613, Egypt

**Keywords:** microbiome, bladder cancer, cystectomy, BCG therapy, TURBT

## Abstract

Bladder cancer is the 10th most common cancer worldwide. Approximately 75% of patients with bladder cancer will present with non-muscle invasive disease. Patients are usually treated with transurethral resection of bladder tumor (TURBT), in addition to adjuvant intravesical therapy (chemotherapy or anti-cancer immunotherapy with Bacillus Calmette Guerin- BCG) for those at intermediate-risk and high-risk of recurrence and progression. For many years, urine has been thought to be “sterile”; however, advanced microbiological and molecular techniques, including 16S ribosomal RNA (16S rRNA) sequencing, have negated that previous paradigm and confirmed the presence of a urinary microbiome. The urinary microbiome has been associated with several urological diseases, including interstitial cystitis, urgency urinary incontinence, neurogenic bladder dysfunction, and others. More recently, many reports are emerging about the role of the urinary microbiome in urothelial carcinogenesis, including gender disparity in bladder cancer and responses to treatments. The urinary microbiome may serve as a biomarker that can help with risk stratification as well as prediction of the response to intravesical therapies. However, the microbiome literature has been hampered by the lack of a unified standardized methodology for sample collection, type, preservation, processing, as well as bioinformatics analysis. Herein we describe and critique the literature on the association between urinary microbiome and bladder cancer and highlight some of the future directions.

## 1. Introduction

Bladder cancer with 573,278 new cases and 212,536 estimated deaths worldwide is the 10th most common malignancy in the world. It is estimated that approximately 81,180 new cases and 17,100 deaths occurred in the United States alone in 2022 [1,2]. Approximately 75% of patients with bladder cancer present with disease that is confined to the mucosa or submucosa (non-muscle invasive bladder cancer [NMIBC]) and 25% present with muscle-invasive disease (MIBC). Patients with NMIBC are further classified into different risk groups (low, intermediate, high and, more recently, highest risk) according to the risk of recurrence and progression, which depends on the T stage, grade, number, size, presence of carcinoma in situ and prior recurrence. For NMIBC, patients are usually treated with transurethral resection of the bladder tumor (TURBT). For patients who are at a higher risk of recurrence and progression, adjuvant intravesical therapy (with chemotherapy or immunotherapy with Bacillus Calmette Guerin- BCG) should be administered [3]. Nevertheless, at 5 years, almost 50% of patients will ultimately experience disease recurrence and 10–30% may progress to MIBC [3,4].

For many years urine has been thought to be “sterile”; however, advanced microbial culture and molecular techniques, including 16S ribosomal RNA (16S rRNA) sequencing, have negated that previous paradigm and confirmed the presence of a very diverse urinary microbiome [5,6,7]. The urinary microbiome may be influenced by many variables, such as gender, infections, smoking status, diet, antibiotic therapy, and many others. While most of the studies that described the urinary microbiome focused on bacteria, the presence of fungi, viruses and archaea have also been described. A recent study showed the presence of a urinary fungal community such as Dothiodeomycetes, Saccharomycetes, Eurotiomycetes, Exobasidiomycetes and Microbotryomycetes [8]. *Candida* spp. has been reported in catheterized urine samples from healthy individuals [9]. The viral community in the urinary tract is mainly composed of bacteriophages, although some eukaryotic viruses have also been described [10].

As mentioned earlier, 16S rRNA gene amplification and sequencing have made it possible to describe the presence of a male urinary microbial community in healthy individuals, as well as a dysbiosis associated with disease. Like other human microbiota, the urinary microbiome could modulate the local immune and inflammatory responses in various urological diseases [11,12]. This might have a profound impact on the course of diseases having inflammation as an underlying factor. The urinary microbiome has been associated with several urological diseases, including interstitial cystitis, urgency urinary incontinence, neurogenic bladder dysfunction, and others [9,13,14]. More recently, many reports are emerging about the role of the urinary microbiome in urothelial carcinogenesis, including gender disparity in bladder cancer [15].

The relationship between micro-organisms and carcinogenesis has been well established for several cancers, including *Helicobacter pylori* and gastric cancer, Human Papilloma Virus and cervical/penile cancers, Epstein Barr Virus with Burkitt lymphoma, Escherichia coli and colorectal cancer, to name a few [16,17,18]. Overall shifts in the human microbiome composition, whether qualitative or quantitative, are shown to cause an alteration of the body’s homeostasis and lead to carcinogenesis [19].

It remains unclear what best defines the urinary microbiome. It could be the taxa residing in the urine, those adherent to the surface of the urothelium or the tumor creating the biofilm, or intracellular taxa, or possibly an interaction between multiple groups (Figure 1). The microbiome is usually described in terms of alpha, beta diversities, and relative abundance. Alpha diversity refers to the diversity of microbial populations within a sample and is estimated by Observed, Chao, Shannon, Simpson, and Ace diversity indices. Beta diversity refers to the differences between microbial populations across different samples and is usually estimated using the Bray–Curtis dissimilarity score paired with classical multi-dimensional scaling. Relative differential abundance measures the balance between specific microbial groups within a community that may be more relevant to the pathogenesis of bladder cancer [20].

In this context, we reviewed the current literature regarding the association between the urinary microbiome and bladder cancer and sought to summarize and critique the currently available evidence. We also described the limitations of the current literature and summarized the future directions.

## 2. Possible Mechanisms: Urinary Microbiome, Inflammation and Bladder Tumor Microenvironment

Microbiota-induced physiological processes impact various key aspects including bioactive agent bioavailability (either by synthesis or uptake), nutrient uptake, immune system development and pathogen displacement, whose alterations have been associated with the development of various diseases including cancer. Chronic inflammation can be attributed to host defense mechanisms against microbial infection or cellular injury in reaction to stressors. However, accumulating evidence suggests that chronic inflammation may play a critical role in various malignancies, including bladder cancer [21], microbial dysbiosis (disturbance of the microbial composition and diversity) and an alteration in the abundance of inflammation-modulating bacteria can modulate inflammatory microenvironment in the bladder that leads to the onset and progression of pathologies like cancer and impacts the cancer treatment options as well [22].

Possible mechanism(s) associated with chronic inflammation during bladder cancer initiation/progression may include sustained inflammatory bladder microenvironment due to an event of microbial dysbiosis. During such an event, bacterial translocation may be enhanced by alterations in the microbiome and host defenses, leading to augmented inflammation. Inflammation may be sustained and regulated by microorganismal pathogen-associated molecular patterns (MAMPs) that activate Toll-like receptors (TLRs) in many cell types to finally activate several signaling pathways linked to the process of bladder carcinogenesis. These inflammation-associated pathways include Janus-activated kinase (JAK)-STAT3, NF-κB, and phosphoinositide-3 kinase (PI3K)-Akt-mammalian target of rapamycin (mTOR). Additionally, the microbiome can directly mediate genotoxic effects by releasing various bacterial genotoxins. During this augmented inflammatory process, reactive oxygen species (ROS) and reactive nitrogen species [23] released from inflammatory cells, as well as hydrogen sulfide (H_2_S) from the microbiota, may also be genotoxic. Also, microbiome metabolism may influence genotoxins such as acetaldehyde, dietary nitrosamines and other carcinogens, hormone metabolism such as estrogen and testosterone and bile acid metabolism that ultimately might as well intensify the proinflammatory bladder microenvironment and lead to more damage. Eventually, carcinogenesis coexists with genotoxicity and inflammation [24,25]. For example, El-Mosalamy et al. specifically concluded that *E. coli* infection might play a role in the development of bladder cancer via activation of NF-κB pathway resulting in inhibition of apoptosis and augmented inflammation [26]. It was further shown by Guo et al. that uropathogenic *E. coli* was shown to induce bladder cancer progression by enhancing bladder tumor angiogenesis via cytotoxic necrotizing factor 1-induced endothelial growth factor expression [27]. This phenomenon has also been observed in non-bladder cancer types. For example, microbes like *Streptococcus gallolyticus*, *Enterococcus faecalis*, Enterotoxigenic *Bacteroides fragilis*, *Escherichia coli*, and *Fusobacterium nucleatum* may contribute to colorectal cancer pathogenesis by triggering inflammation and DNA damage [28].

It should also be kept in mind that the presence of some microbes can also attenuate inflammation in a context-dependent manner. For example, studies have shown that some resident commensal and probiotic bacteria attenuate mucosal inflammation by downregulating the NF-κB pathway, IL-6 and IL-8 [29]. This scenario, on one hand, could limit urinary tract infections in the healthy; however, it could impair responses to beneficial BCG therapy on the other hand. Adding more hurdles to the latter, the urinary microbiome may influence the possible response to BCG therapy via BCG destruction/inactivation in bladder lumens or by limiting urothelial sensitivity to BCG activity by competitively binding to fibronectin in the presence of BCG [30]. Therefore, identifying the bacterial species involved (with their mechanism of action known) could be beneficial to understanding and preventing the onset and treatment of tumors and/or controlling their advancement and progression.

While the involvement of inflammation in tumor formation including that of the bladder is becoming increasingly clear, interference with the inflammatory tumor microenvironment has been shown to inhibit anti-tumor activity. Therefore, it is critical to characterize the link between the urinary microbiome and chronic inflammation in the bladder which might be crucial to enable the development of novel strategies for bladder cancer prevention and treatment. Further, such novel strategies must not compromise the anti-tumor efficacy of immunotherapies like BCG or immune check-point immunotherapies.

## 3. Literature Review

### 3.1. Bladder Cancer versus Controls

Hussein et al. found no significant difference in alpha diversity but found a significant difference in beta diversity at the genus level [31]. On the other hand, Oresta et al. found a significantly higher alpha diversity evenness index in bladder cancer patients, but there was no significant difference in beta diversity [32]. Moynihan et al. as well as Bucevic Popovic et al. found no significant difference in alpha or beta diversity [33,34].

A higher abundance of phyla Actinobacteria and Proteobacteria was observed in the urine of patients with bladder cancer [31,35,36], while Firmicutes was higher in controls [31,36]. Deinococcus-Thermus was higher in bladder cancer in one study [31] and higher in controls in another [36].

At the genus level, Hussein et al. reported a higher abundance of Actinomyces, Achromobacter, Brevibacterium, and Brucella in the urine samples of bladder cancer patients [31]. For Corynebacterium, Oresta et al. [32] and Moynihan et al. [33] reported increased relative abundance in bladder cancer patients, while Bucevic Popovic et al. and Pederozli et al. reported decreased relative abundance. Another frequently reported pathogen was Acinetobacter; three studies [36,37,38] reported increased abundance in bladder cancer patients, while Pederozli et al. showed a decreased abundance in patients with bladder cancer [39]. Other uropathogens that were more abundant in bladder cancer patients included Klebsiella [39], Esherichia-Shigella [36], Brucella [31], and Pseudomonas [40].

When stratified according to the type of sample, Acinetobacter, Aeromonas, Actinomyces were increased in mid-stream urine of bladder cancer patients [32,37,41,42]. On the other hand, Lactobacillus and Veillonella were higher in the mid-stream urine of controls [35]. When considering studies that used catheterized urine, Veillonella had a higher relative abundance in the bladder cancer group [32]. On the other hand, when considering studies that used tissue sample analysis, Lactobacillus was more abundant in the control groups [31,35] (Table 1).

### 3.2. Males versus Females with Bladder Cancer

Several studies investigated gender differences in the microbiome. Higher alpha diversity was observed in tissue specimens of male patients but there was no significant gender difference in urine samples [42]. Another recent study showed no difference in the alpha or beta diversities in urine samples between males and females [31].

In terms of differential abundance at the phylum level, female patients demonstrated a higher differential abundance of several phyla, mainly Bacteriodetes, while male patients demonstrated a higher relative abundance of Actinobacteria. At the genus level, female patients demonstrated a higher abundance of 21 bacterial genera including Lactobacillus, Actinotignum, Prevotella, Vellionella, Campylobacter, and Enterococcus. It is noteworthy that although the most abundant genus found in a healthy women’s urinary microbiome is *Lactobacillus*; however, not all *Lactobacillus* species are associated with a healthy microbiota [13,43,44]. While *Lactobacillus crispatus* has been associated with a healthy state, *Lactobacillus gasseri* has been described in disease states such as urgent urinary incontinence [9]. Also, even a diminished abundance of *Lactobacillus* can lead to a pathologic state, as the lower abundance may favor the colonization of disease-causing uropathogens [13].

For males, Pelomonas, Corynebacterium, Finegoldia, and hgcl clade were more abundant [31]. Other studies reported Actinotigum (higher in females) [31,42], as well as Streptococcus (higher in males) [42].

### 3.3. Bladder Cancer Stage/Severity

Few studies have focused on the exact differences in taxa based on disease stage. Hussein et al. observed no significant difference in alpha diversity or beta diversity indices comparing urine samples from NMIBC with those with MIBC [31], but Zeng et al. observed a higher Shannon and lower Simpson indices in patients with bladder cancer who developed recurrence [41].

*Haemophilus* and *Veillonella* were significantly increased in MIBC [31], while *Cupriavidus* [31], *Staphylococcus* [37], *Campylobacter* [34], and *Corynebacterium* [32] were more abundant in NMIBC (Table 1). *Escherichia_Shigella* was reported to be in higher abundance in NMIBC with a lower risk of recurrence [37].

### 3.4. BCG Responsiveness

Few studies investigated differences based on BCG responsiveness. There was no difference in alpha or beta diversity between responders and non-responders. In terms of differential abundance, *Serratia*, *Pseudomonas*, *Brochothrix*, and *Negativicoccus* at the genus level were found significantly more abundant in BCG-responders [31] (Table 1).

### 3.5. Urethral Microbiota

One study examining the difference between urine samples obtained by cystoscopy and voided urine showed that there are differences in beta diversity in males while not in females, attributing that to longer urethra length in males [45]. Oresta et al. investigated the differences between voided and catheterized urine to determine the urethral microbiome. For instance, they found that Fusobacterium was unique to the voided samples [32]. The authors also suggested that mid-stream urine is subject to contamination by opportunistic taxa like Streptococcus, Corynebacterium, and Enterococcus. Although mid-stream urine remains likely the most convenient way to provide samples from both patients and normal subjects, it is subject to contamination by urethral or vaginal microbiota.

### 3.6. Bladder Cancer Tissue

Authors mostly utilized urine for studying the microbiome with only four studies that examined the cancer tissue. Liu et al. showed that bladder cancer tissue had significantly less microbial richness in terms of both alpha and beta diversities, while Mansour et al. did not find any difference in diversity compared to controls [36,42]. At the genus level, two of these studies showed an increase in Lactobacillus in the control groups [36,42]. Many phyla and genera were higher in the bladder cancer group, with no overlap between any of the studies (Table 2).

## 4. Discussion

Bladder cancer poses a significant burden on healthcare costs and confers a significant impact on the life expectancy of afflicted patients. However, its etiology and carcinogenesis are yet to be fully elucidated [1]. The urinary microbiome may be implicated in urothelial carcinogenesis via different mechanisms. Commensal bacteria inhabit mucosal surfaces and interact with the immune system to form a physiological immune barrier [23]. If these barriers are infiltrated by opportunistic bacteria, they may trigger pro-inflammatory pathways that may lead to carcinogenesis by DNA damage. For example, *E. coli* has been known to produce the genotoxin colibactin, which leads to DNA damage in colorectal cancer [48]. The urinary microbiome may serve several functions, including maintenance of homeostasis of the urinary tract by development and regulation of uroepithelial junctions [49]. Waste products produced by the detoxification of products of smoking and occupational exposure are filtered by the kidneys and temporarily stored in the bladder, where they can affect the microbial composition of the urine. Nadler et al. showed that some bladder tissue exhibited bacterial biofilms in patients with bladder cancer and the authors concluded accordingly that microbial biofilms may play a role in the carcinogenesis of bladder cancer, but future investigations are needed to quantify this finding and speciate the bacteria present. The concept of oncogenic potential of dysbiotic microbiome biofilms has been studied in colorectal cancer. We believe that this concept can be extrapolated to bladder cancer as well [50].

The role of bacterial urinary tract infections in bladder cancer incidence has been studied at an epidemiological level. However, the data is conflicting. For instance, some of the studies report increased rates of bladder cancer with more urinary tract infections [51]. Other studies show that repeated urinary infections confer a significant reduction in the risk of bladder cancer development in women [52]. More recently, a systematic review of recent articles found no association between bladder cancer and urinary tract infection [53]. However, the intertwined role of bacteria and bladder cancer has been there for a long time. The attenuated BCG vaccine has been used to treat NMIBC for decades, likely by inducing an anti-tumor immunological response to eliminate cancer cells, although the exact mechanisms remain poorly understood [3]. Moreover, probiotic intake has been shown to reduce the probability of recurrence after radical cystectomy, which enforces the concept of the microbiological environment influencing cancerous cells [54].

Differences in the urinary microbiome can also explain the difference in bladder cancer prevalence between men and women. *Lactoboacillus*, a commonly found microorganism in the genitourinary microbiome of women, might explain the gender disparity in bladder cancer in addition to other factors such as smoking and occupational exposure, possibly by reducing inflammation, as previously proposed [55]. Fermented dairy product has been previously shown to possibly decrease bladder cancer risk, which might be attributed to the effect of *Lactobacillus* [56]. It has been shown that *Lactobacillus* probiotics can decrease NMIBC recurrence [57,58].

The urinary microbiome influences the local bladder immune microenvironment and influences innate and adaptive immunity [59]. There is wide variation in response to BCG therapy from one patient to another with NMIBC. The exact mechanism of how BCG works remains to be fully elucidated. Few mechanisms were proposed, including BCG binding to fibronectin with subsequent induction of CD8^+^ T and natural killer cell responses [60,61]. It may also induce crosslinking between α5β1 integrins with subsequent cell cycle arrest, or generation of reactive oxygen species [62]. There is lack of a complete understanding on how resistance to BCG develops. One of the postulations is the effect of the individual’s microbiome and its interaction with BCG and urothelium, either by promoting or blocking the effects of BCG mycobacteria. Some bacteria can bind fibronectin and attenuate mucosal inflammation by attenuation of the NF-kB pathway, therefore either improving or decreasing the efficacy of BCG [49]. The urinary microbiome could modulate the response to BCG immunotherapy by competitively binding to cellular components, such as fibronectin and α5β1 integrins, essential for BCG function [63]. Therefore, specific commensal bacteria may saturate these binding sites, and thus could decrease BCG efficacy and potentially downregulate the strong cytotoxic response needed to eliminate tumor cells [64]. For example, *Lactobacillus iners*, found in the urinary microbiome, binds to fibronectin with higher affinity than any other species [65]. This may suggest that *Lactobacillus* might synergize with BCG to amplify the elicited response to treatment; however, further work needs to be done to determine the exact mechanism by which the urinary microbiome could modulate BCG immunotherapy.

In this context, the urinary microbial profile can possibly be utilized to predict the response to intravesical BCG, help with risk stratification and therefore allows early referral of patients who are unlikely going to respond to BCG for early cystectomy, or at least consider other options for BCG refractory disease. Another potential benefit is modulation of the urinary microbiome, for instance, using probiotic supplements to potentiate the effect of intravesical BCG. There is an ongoing clinical trial (SILENTEMPIRE) to investigate the use of microbial profiles from the bladder and the feces of NMIBC patients as a predicting tool for therapy response prior to BCG administration. In this trial, patients’ urinary and fecal microbiomes are collected prior to the initiation of BCG therapy. The relationship between the microbial profile and BCG response will then be determined [66]. Sweis et al., while characterizing the role of the urinary microbiome in patients with high-risk NMIBC undergoing BCG treatment, demonstrated that the abundance of *Proteobacteria* like *Gammaproteobacteria* was associated with recurrence, while Firmicutes such as *Lactobacillales* were more abundant in patients without recurrence [67]. Consequently, probiotics may prove beneficial in the treatment of bladder cancer as studies showed that participants who consumed fermented milk products and probiotics achieved a reduction in bladder cancer incidence and recurrence [54,68].

## 5. Limitations of the Current Literature

The urinary microbiome literature has been hampered by variations in sample size, the inclusion of female patients, diversity assessments, DNA extraction methods, sequencing libraries, and bioinformatics analyses. Moreover, none of the studies reported abundance at the species level. Microbiota biodiversity profiling in bladder cancer is still in the beginning stages. There are still multiple hurdles the scientific community must overcome to better elucidate the qualitative role of the microbiome in bladder cancer as well as quantitate the type of bacterial species responsible for carcinogenesis. Methods of urine sample collection also significantly impact the study of the urinary microbiome. Urine samples can be collected using various methods including the collection of spontaneous midstream urine, catheterization with an intermittent or permanent catheter or suprapubic aspiration from the bladder. Many studies have used mid-stream urine, while other studies used catheterized urine or urine recovered during cystoscopy. On the other hand, there are some studies who studied the tissues for the microbial diversity. While Mansour et al. reported that the urinary microbiome can adequately represent the bladder cancer tissue microenvironment, they found a significant difference between the tissue microenvironment and catheterized urinary microbiome in female patients, owing to that difference to genital floral contamination [42]. Oresta et al. revealed a significantly different abundance of micro-organisms between mid-stream urine collections and catheterized urine [32]. The authors suggest that mid-stream urine is subject to contamination by opportunistic taxa like *Streptococcus*, *Corynebacterium*, and *Enterococcus*. These bacteria would not reflect the pathophysiology of the cross-talk between the true bladder microbiome and the urothelium lining it. Another confounding factor to impact urinary microbiome is age. While Curtiss et al. evaluated the microbiome of 79 healthy women to identify changes related to age and menopausal status, they observed a greater incidence of *Lactobacillus* in the urinary microbiome of pre-menopausal women compared to post-menopausal women and vice versa for *Mobiluncus* [69]. Possible mechanisms for this disparity might include declining levels of estrogen during menopause that induces vulvovaginal atrophy to impair the defense against invading pathogens and contribute to the increased risk for urinary tract infections. Additionally, incomplete emptying of the urinary bladder after voiding also might increase the risk of recurrent urinary tract infections as residual urine and decreased urine flow in the absence of estrogen impairs the mechanical clearance of bacteria, thereby causing pathogens to colonize the bladder [70]. Such a mechanism might explain the abundance of genera including *Jonquetella*, *Parvimonas*, *Proteiniphilum*, and *Saccharofermentans* that appear exclusively in the urinary microbiome of older individuals [71]. Eto et al. have shown that strains of uropathogenic *E. coli* can invade urothelial cells causing the redistribution of actin microfilaments anddevelopment of pod-like inclusions or biofilm, which can be linked to recurrent urinary tract infections. It remains unclear how the urinary microbiome interacts with biofilms and how this can affect the occurrence of bladder cancer [72]. In addition to age, non-modifiable host factors like gender and genetics may influence the innate immune responses and thus could influence the type of bacterial colonization.

All the studies used 16S rRNA sequencing amplification for their analysis with significant variation in the sub-region of amplification. Most of the studies amplified V3 and V4 or a combination of the two sub-regions. The studies had a variety of sequencing and pipelines that were used, whereby most studies used MISeq and HISeq sequencing platforms and QIIME and UPARSE pipelines. Furthermore, reference databases were varied (SILVA, Greengenes). As for the diversity assessments, different diversity indices were used for alpha diversity (ACE, Shannon, Simpson, Chao1), while fewer studies assessed beta diversity, and the estimates included component and coordinate analysis as well as UniFrac distances among the studies tackling different method of urine collection. This may explain, in part, the wide variation of taxa described in different studies.

Another hurdle worth mentioning is the concern of the best control patients. Current data is unclear as to which samples would be ideal controls for comparison. Some studies utilized normal-looking bladder mucosa from known patients with bladder cancer as controls. This approach may not be ideal for several reasons. First, bladder cancer can be multifocal, and, therefore, the mucosal abnormalities may be present in multiple areas within the bladder even without a visible tumor. Second, it has been shown in colorectal cancer literature that the extent of abnormal microbiome extends beyond the area where the cancer is. Even when using controls without any GU cancers, several limitations exist. There are many variables that may change the microenvironment of the bladder; controls can be subject to unknown biases. For example, previous antibiotic intake might change the microenvironment, glucosuria is another variable that might alter the microenvironment, other variables might also include gender, age, probiotic intake, and others. Another crucial point is that the microbiome may change over time and with different environmental exposures and other lifestyle habits (such as smoking, source of drinking water, etc.). Assessment of the microbiome at a single time point may not really capture the full spectrum. In summary, data remains inconsistent between the different studies, and the potential role of uropathogens in bladder cancer is still not well understood.

## 6. Future Directions

The potential benefits of the urinary microbiome in bladder cancer are multifold. Microbiomes can modulate the response of BCG in NMIBC by amplifying the immune response of BCG or attenuating it. Microbial species have been added to increase the efficacy of nivolumab/ipilimumab in metastatic RCC in a recent phase one trial [73]. Another possibility is to stratify patients to the best therapeutic pathway after resection of the primary tumor whereby the microbiome of the tumor would be an indicator of chemosensitivity or sensitivity to immunotherapy.

Future investigations should also focus on evaluating the roles and associated mechanisms of microbiota that strongly associate with recurrence vs. no recurrence post BCG-treated NMIBC disease. The immunobiology of those microbiotas that associate with no recurrence post-treatment should be thoroughly studied so that mechanisms of anti-tumor immune re-activation or modulation can be deciphered, tested and finally utilized for bladder tumor control and treatment strategies. This strategy would be ideal to identify mechanisms of bladder cancer control through the use of the urinary microbiome.

The microbiome likely modulates the body’s immune system and anti-neoplastic properties that may impact bladder cancer progression and tumorigenesis. Therefore, the microbiome might be utilized as a biomarker as well as in the management of bladder cancer at different stages of the disease. More importantly, the complexity and heterogeneity of the derived microbial data in the literature, a unified methodology in sampling as well as processing and computational techniques are required to incorporate clinical, biological, and microbiological variables.

It is also crucial to study other components of the urinary microbiome, including fungi and viruses. Another critical point is whether isolated taxa from bladder cancer patients represent the actual taxa associated with bladder cancer, or is it the disappearance of the protective taxa that is associated with the disease onset? Interaction with other components of the immune system can add another layer of complexity to the study of the urinary microbiome.

## Figures and Tables

**Figure 1 life-13-00812-f001:**
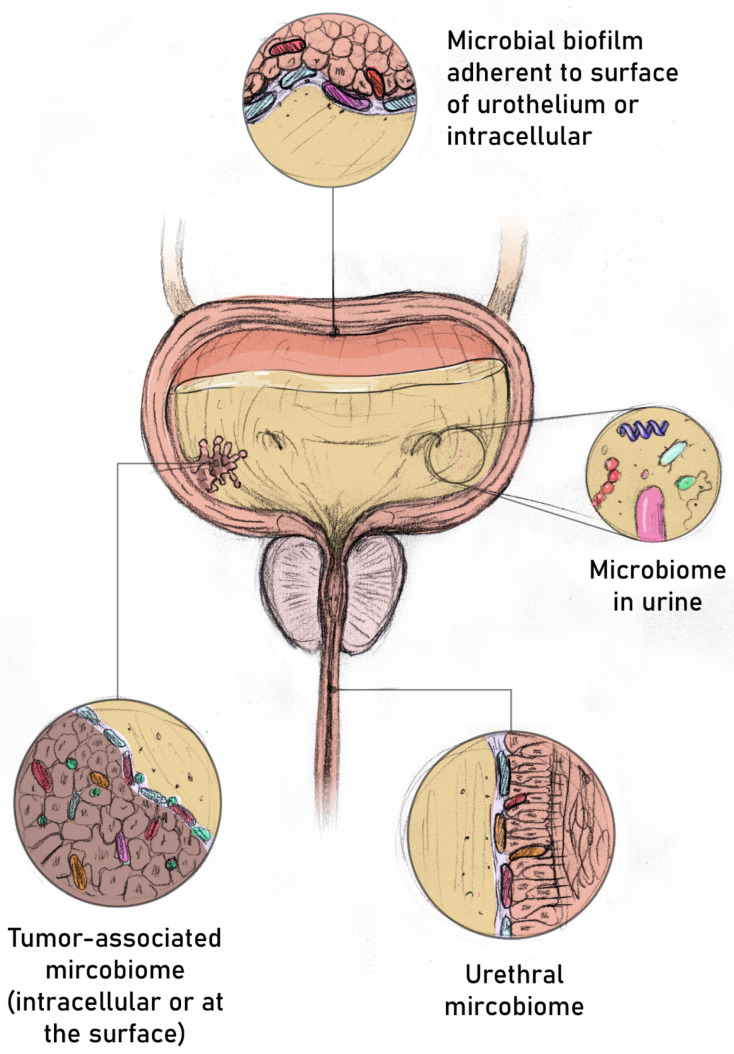
Multiple sites of taxa that may be contributing to the urinary microbiome.

**Table 1 life-13-00812-t001:** Studies of the urinary microbiome in bladder cancer that have used urine samples.

Study, Year	BCa, n	Non-BCa, n	Females n (%)	Sample	Alpha-Diversity	Beta-Diversity	Relative Abundance, Phyla	Relative Abundance, Genera	Other
Xu et al., 2014 [40]	8	6	NA	Urine—clean catch	Not reported	Not reported	**BCa:** Streptococcus **Non-BCa:** Streptococcus abundance near 0	N/A	--
Wu et al., 2018 [37]	31	18	0	Urine—clean catch	Different	Different	**BCa:** Proteobacteria Firmicutes Actinobacteria **Non-BCa:** Proteobacteria Firmicutes Bacteroidetes	**BCa:** Escherichia-Shigella Staphylococcus Streptococcus Aeromonas **Non-BCa:** Escherichia-Shigella Staphylococcus Streptococcus Lactobacillus	--
Bucevic Popovic et al., 2018 [34]	12	11		Urine—clean catch	NS	NS	Firmicutes, Actinobacteria, Bacteroidetes and Proteobacteria	Streptococcus, Prevotella, Peptoniphilus, Campylobacter, Veillonella, Anaerococcus, Finegoldia	
Mai et al., 2019 [38]	24	0	6 (25%)	Urine—clean catch	Not reported	Not reported	The five most abundant phyla are Proteobacteria, Firmicutes, Actinobacteria, Tenericutes, and Bacteroidetes	There are 31 bacterial genera (Core31) in all of these 24 samples, including Clostridiales_f_g, Peptoniphilus, Mycoplasma, Cupriavidus, Lachnospiraceae, Ureaplasma, Delftia, o_Rhizobiales_f_g, Acinetobacter, Enterococcus, Hydrogenophilus, Prevotella, Bacillus	
Bi et al., 2019 [35]	29	26	20 (36%)	Urine—clean catch	different	different	Tenericutes and Proteobacteria was higher in the cancer group versus the control group	Non-BCa: Streptococcus, Bifidobacterium, Lactobacillus and Veillonella BCa: Actinomyces	
Moynihan et al., 2019 [33]	8	33	0	Mid-stream urine	NS	NS	Firmicutes, Proteobacteria, and Bacteroides	Turicibacter, Lactobacillus, and Bacteroides	
Hourigan et al., 2020 [45]	22	0	8	Cystoscopy and mid-stream urine	NS	NS	Firmicutes (increased in voided samples) and Proteobacteria (increased in males)	Sternotrophomonas (increased in cystoscopy samples) and tepidimonas (increased in males)	
Ishaq et al., 2020 [46]	10	0	3	Midstream urine and tissue	NS	NS		Enterobacteriaceae, Bacillus, Meiothermus and Methylotenera	
Chipollini et al., 2020 [47]	27	10		Mid-stream urine	Different	Different	NA	NA	
Mansour et al., 2020 [42]	10	0	5	Mid-stream urine and tissue	NS	NS	In urine samples, the most abundant phyla detected were Firmicutes, Proteobacteria, Actinobacteria, Cyanobacteria, and Bacteroidetes. In the tissue samples, Firmicutes, Actinobacteria, Proteobacteria, Bacteroidetes, Cyanobacteria	urine samples: Lactobacillus, Corynebacterium, Streptococcus and Staphylococcus. Tissue samples: Bacteroides, Akkermansia, Klebsiella and Clostridium	Akkermansia, Bacteroides, Clostridium sensu stricto, Enterobacter and Klebsiella genera showed remarkably higher compositional abundance in tissue than in urine samples
Pederzoli et al., 2020 [39]	49	59	38	Mid-stream urine Tissue	NS	In urine: Different In tissue: NS	Proteobacteria, Firmicutes, and Bacteroidetes		Sought to find shared BCa microbiome between urine and bladder tissue
Zeng et al., 2020 [41]	62	19	0	Mid-strean urine	Different	Different in males only	Firmicutes, Proteobacteria, and Actinobacteria	Staphylococcus, Streptococcus, Prevotella, and Corynebacterium	
Hussein et al., 2021 [31]	43	10	12	Mid-stream urine (healthy) Cath/cystoscopy from cancer patients	NS	Different	BCa: Actinobacteria, Proteobacteria Non- BCa: Firmicutes, Deinococcus-Thermus	BCa: Actinomyces, Achromobacter, Brevibacterium, and Brucella Non-BCa: Salinococcus, Jeotgalicoccus, Escherichia-Shigella, Faecalibacterium, Thermus, and Lactobacillus	MIBC vs. NMIBC BCG responders vs. non responders Male vs. females with BCa
Oresta et al., 2021 [32]	51	10	0	Mid-stream urine and catheterized urine	Different	NS	Firmicutes, Actinobacteria, Bacteroidetes and Pro-teobacteria	Veillonella and Corynebacterium were enriched in the BC group	Veillonella was increased in pTa/T1 HG, CIS and T2 tumors compared to controls and pTa LG tumors; Corynebacterium and Staphylococcus were specifically enriched in HG NMIBC and pTa LG tumors, respectively

**Table 2 life-13-00812-t002:** Studies of urinary microbiome in bladder cancer that have used tissue samples.

Study, Year	BCa, n	Non-BCa, n	Females n (%)	Sample	Alpha-Diversity	Beta-Diversity	Relative Abundance, Phyla	Relative Abundance, Genera	Other
Liu et al., 2019 [36]	22	12	0	BCa and non-BCa tissue	NS (except for Shannon Index)	Different	BCa: Proteobacteria, Actinobacteria, Cyanobacteria, Chloroflexi, Deinococcus-Thermus, Armatimonadetes Non- BCa: Firmicutes, Bacteroidetes	BCa: Cupriavidus, Uncl Brucellaceae, Acinebacter, Escherichia-Shigella, Sphingomonas, Pelomonas, Ralstonia, Anoxybacillus Non-BCa: lactobacillus, Prevotella, Ruminococcaceae	High Grade vs. Low Grade High risk for recurrence and progression
Ishaq et al., 2020 [46]	10	0	3	Midstream urine and tissue	NS	NS		Enterobacteriaceae, Bacillus, Meiothermus and Methylotenera	
Mansour et al., 2020 [42]	10	0	5	Mid-stream urine and tissue	NS	NS	In urine samples, the most abundant phyla detected were Firmicutes, Proteobacteria, Actinobacteria, Cyanobacteria, and Bacteroidetes. In the tissue samples, Firmicutes, Actinobacteria, Proteobacteria, Bacteroidetes, Cyanobacteria	urine samples: Lactobacillus, Corynebacterium, Streptococcus and Staphylococcus. Tissue samples: Bacteroides, Akkermansia, Klebsiella and Clostridium	Akkermansia, Bacteroides, Clostridium sensu stricto, Enterobacter and Klebsiella genera showed remarkably higher compositional abundance in tissue than in urine samples
Pederzoli et al., 2020 [39]	49	59	38	Mid-stream urine Tissue	NS	In urine: Different In tissue: NS	Proteobacteria, Firmicutes, and Bacteroidetes		Sought to find shared BCa microbiome between urine and bladder tissue

## Data Availability

Not applicable.

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
