# Peer review of "The Urinary Microbiome and Bladder Cancer"

_life, 2023, doi:10.3390/life13030812_

Round 1
Reviewer 1 Report
Nassib et al. reviewed the literature to explain the relationship between urinary microbiome and bladder cancer. This manuscript is well-written, and could help the readers to understand the current research hot topics in this field. I would recommend its publication, after some minor revisions.
1. Since the alpha and beta diversities are important parameters in related research, a straightforward and brief graphical illustration could be helpful.
2. I noticed that the authors emphasized gender differences in the microbiome research. But for other patients' characteristics, such as smoking status, age, and occupational exposure, are there any reports to determine whether and how these confounding factors influence the relationship between urinary microbiome and bladder cancer?
Author Response
We would like to thank the reviewer for their constructive comments. Please find our response below.
- Since the alpha and beta diversities are important parameters in related research, a straightforward and brief graphical illustration could be helpful
-The microbiome literature is usually described in alpha, beta diversities and differential abundance. This was explained in detail in the manuscript alongside the commonly used tests to evaluate them (Lines 70-76).
I noticed that the authors emphasized gender differences in the microbiome research. But for other patients' characteristics, such as smoking status, age, and occupational exposure, are there any reports to determine whether and how these confounding factors influence the relationship between urinary microbiome and bladder cancer?
We agree with the reviewer. These confounding variables can influence the urinary microbiome, and unfortunately represent a major limitation to the microbiome literature. Some studies tried to account for these variables by including them in the multivariate analysis similar to a previous study published by our group in the Journal of Urologic Oncology (reference number 31). To our knowledge, no studies specifically looked into smoking, age and occupational exposure. This was highlighted in the "limitations of the current literature" section (Lines 369-373).
Reviewer 2 Report
The present paper provides interesting insight into the association between urinary microbiome and bladder cancer. The ambition and conclusions of the paper are fitted with the scope of the journal. The review is relevant and interesting, and conclusions consistent with the evidence and arguments presented.
The title accurately reflects the content and, in general, the abstract presents an adequate synopsis of the paper. The introduction section provides a good, generalized background of the topic with logically organized, clear and well-argued narrative.
Language: The English in the present manuscript requires minor improvement –in certain passages there are certain inconsistencies regarding grammar/vocabulary and sentence structure.
The main study-review part requires better structural organization
Line 142: Please, re-write the subsection title in order to improve clarity
Line 155: Please, correct the Popovic et al typo (author’s surname)
Line 188: Please check the subsection numbering (the title features number 5 albeit 2, 3 and 4 are missing)
Author Response
We would like to thank the reviewer for their constructive comments. Please find our response below.
Language: The English in the present manuscript requires minor improvement –in certain passages there are certain inconsistencies regarding grammar/vocabulary and sentence structure. The main study-review part requires better structural organization.
Thank you for the comment. The manuscript was revised and the section heads were changed to improve readability.
Line 142: Please, re-write the subsection title in order to improve clarity
Line 155: Please, correct the Popovic et al typo (author’s surname)
Line 188: Please check the subsection numbering (the title features number 5 albeit 2, 3 and 4 are missing)
The manuscript was revised and structural typos, author name referenced in the article were updated.
Reviewer 3 Report
The manuscript by Heidar et al. (The Urinary Microbiome and Bladder Cancer) is interesting because the subject is currently controversial. The authors make an honest presentation of the state of knowledge.
However, the form of the manuscript could be improved to make it more pleasant, especially the tables which can be more concise. On the other hand, the figure is very synthetic even if the legend is minimalist.
We think that three small sections could be added in order to strengthen the manuscript. A better detailing of bladder cancer and its progression. There is only a single short paragraph for now. Also a better description of bacterial infection processes would be welcome.
This leads us naturally to a third paragraph which could propose a connection between the modifications of the cytoskeleton during the bacterial infection and in particular the recurrent infection with E.Coli (Cellular Microbiology (2006)8(4), 704–717) and the epithelial-mesenchymal transition. In particular, whether this disorganization would have beneficial effects or, on the contrary, could lead to tumor progression (Frontiers in Bioscience 7, e1-8, January 1, 2002; Biochem Biophys Res Commun. 2017 Dec 9;494(1-2):165-172.).
Author Response
We would like to thank the reviewer for their constructive comments. Please find our response below.
-However, the form of the manuscript could be improved to make it more pleasant, especially the tables which can be more concise. On the other hand, the figure is very synthetic even if the legend is minimalist.
-It is unclear what the reviewer meant by the figure is very synthetic even if the legend is minimalist. We are happy to update the figure or the legend if needed.
We think that three small sections could be added in order to strengthen the manuscript. A better detailing of bladder cancer and its progression. There is only a single short paragraph for now. Also a better description of bacterial infection processes would be welcome.
This leads us naturally to a third paragraph which could propose a connection between the modifications of the cytoskeleton during the bacterial infection and in particular the recurrent infection with E.Coli (Cellular Microbiology (2006)8(4), 704–717) and the epithelial-mesenchymal transition. In particular, whether this disorganization would have beneficial effects or, on the contrary, could lead to tumor progression (Frontiers in Bioscience 7, e1-8, January 1, 2002; Biochem Biophys Res Commun. 2017 Dec 9;494(1-2):165-172.).
-We agree with the reviewer. Lines 33-36 were added about bladder cancer pathology and Lines 341-345 to discuss the relationship between UTIs, interaction with cytoskeleton, formation of biofilm and the unclear relationship with microbiome and bladder cancer.